# The Roles of Extracellular Vesicles in the Progression of Renal Cell Carcinoma and Their Potential for Future Clinical Application

**DOI:** 10.3390/nano13101611

**Published:** 2023-05-11

**Authors:** Masashi Takeda, Shusuke Akamatsu, Yuki Kita, Takayuki Goto, Takashi Kobayashi

**Affiliations:** 1Department of Urology, Graduate School of Medicine, Kyoto University, Kyoto 606-8507, Japan; m_takeda@kuhp.kyoto-u.ac.jp (M.T.); kitayuki@kuhp.kyoto-u.ac.jp (Y.K.); goto@kuhp.kyoto-u.ac.jp (T.G.); 2Department of Urology, Graduate School of Medicine, Nagoya University, Nagoya 466-8550, Japan; akamats@med.nagoya-u.ac.jp

**Keywords:** extracellular vesicle, renal cell carcinoma, biomarker, therapeutics, cancer progression

## Abstract

Renal cell carcinoma (RCC) is the most common type of kidney cancer and is thought to originate from renal tubular epithelial cells. Extracellular vesicles (EVs) are nanosized lipid bilayer vesicles that are secreted into extracellular spaces by nearly all cell types, including cancer cells and non-cancerous cells. EVs are involved in multiple steps of RCC progression, such as local invasion, host immune modulation, drug resistance, and metastasis. Therefore, EVs secreted from RCC are attracting rapidly increasing attention from researchers. In this review, we highlight the mechanism by which RCC-derived EVs lead to disease progression as well as the potential and challenges related to the clinical implications of EV-based diagnostics and therapeutics.

## 1. Introduction

Extracellular vesicles (EVs) are nanosized lipid bilayer vesicles that are secreted into extracellular space by nearly all cell types, including cancer cells and non-cancerous cells [1,2]. When EVs were first discovered, they were believed to be a system for excreting cellular wastes [3,4]. Since the discovery by Valadi et al. that EVs carry RNA with the ability to function as an intercellular messenger, there has been a rapid increase in the number of publications supporting the crucial role of this novel type of mediator in both physiology and pathophysiology [5,6,7]. EVs mediate various intercellular communication events through the function of their cargo, which can include DNA, RNA, proteins, and lipids, leading to phenotypic changes in recipient cells [8,9,10,11]. In the field of cancer research, accumulating evidence suggests the contribution of EVs to cancer development, invasion, metastasis, and drug resistance by mediating interactions between cancer cells and cells residing in the tumor microenvironment (TME) [12,13,14,15,16,17]. Recently, tremendous efforts have been made to elucidate the roles of EVs in cancer progression, leading to the development of novel technologies for the early detection and treatment of cancer [18,19].

According to the global cancer statistics published in 2020, an estimated 430,000 patients were newly diagnosed with kidney cancer, and 170,000 patients died from kidney cancer [20]. In total, 25–30% of kidney cancer patients present with metastatic disease at the initial diagnosis, and 20–30% of patients with localized disease experience recurrence in distant organs after treatment with curative intent [21,22]. Localized kidney cancer is typically treated with surgery, whereas metastatic kidney cancer is typically treated with systemic therapy [23]. Renal cell carcinoma (RCC) is a cancer that is thought to originate from renal tubular epithelial cells and accounts for about 90% of kidney cancer [24]. The major histological subtypes of RCC include clear-cell renal cell carcinoma (ccRCC), papillary renal cell carcinoma, and chromophobe renal cell carcinoma. Of these, ccRCC is the most frequent and accounts for 90% of all RCC histology types. Minor histological subtypes include collecting duct carcinoma, renal medullary carcinoma, MiT family translocation RCCs, and unclassified RCCs [24]. While localized RCC is associated with an acceptable clinical outcome, with a 5-year survival rate of 93%, metastatic RCC is associated with poor clinical outcome, with a 5-year survival rate of 12% [25]. Thus, further research should focus on the molecular mechanism underlying RCC progression to contribute to developing new methods for early detection and treatment options for RCC.

In this review, we highlight the mechanism by which RCC-derived EVs lead to disease progression and the potential and challenges related to the clinical implications of EV-based diagnostics and therapeutics.

## 2. Overview of EVs

### 2.1. Classification of EVs

Extracellular vesicles (EVs) are lipid bilayer vesicles that are secreted from cells into the extracellular spaces. EVs are heterogeneous and can be classified into three distinct types based on their size and biogenesis: exosomes, microvesicles, and apoptotic bodies [26]. Exosomes are a subpopulation of EVs with diameters ranging from 30 to 150 nm [18,27,28]. Exosomes are generated by the fusion of a multivesicular body (MVB), a late endosomal organelle, with the plasma membrane [29]. Tetraspanins, including CD9, CD63, and CD81, heat shock proteins, and MVB-associated proteins such as Alix and TSG101 are well known to be abundant in exosomes [30,31,32]. Exosomes are the most studied subtype of extracellular vesicles (EVs) due to considerable evidence supporting their roles in numerous pathological processes, including cancer progression [33,34,35]. Microvesicles (MVs) are defined as EVs with diameters ranging from 100 to 1000 nm that bud directly from the plasma membrane into the extracellular space [36]. Many studies have shown that MVs are crucial in cancer progression [25,37]. Several proteins such as integrins, selectins, and CD40 are reported to be concentrated in MVs [38]. Apoptotic bodies are secreted from apoptotic cells and vesicles with diameters ranging from 500 to 5000 nm. The biological process for the generation and secretion of an apoptotic body has not yet been fully elucidated, and further study is required [39]. In addition to the three major types of EVs, there are various terms for membrane vesicles, e.g., ectosomes, oncosomes, shedding vesicles, microparticles, and prostasomes [40]. Therefore, the International Society for Extracellular Vesicles (ISEV) recommends using the term “extracellular vesicle (EV)” as the “generic term for particles naturally released from the cell that are delimited by a lipid bilayer and cannot replicate” and classifying EVs into small and large EVs [41].

### 2.2. EV Isolation and Characterization Methods

There are five categories of EV isolation techniques that are commonly used in laboratories: ultracentrifugation (UC), size-based methods such as filtration and size-exclusion chromatography (SEC), immunoaffinity-based capture, precipitation, and microfluid-based isolation methods [42,43]. Of these, differential centrifugation involving UC is the most traditional and widely accepted method and employed in EV isolation from various sample types, such as biological fluids and cell culture supernatants. Protocols vary based on researcher and sample type; however, sequential centrifugations and filtration through 0.22 mm membranes are typically performed to remove cell debris and cellular organelles before two rounds of UC at 100,000–120,000× *g* for 60–90 min [42,44,45,46]. After UC, pellets containing EVs are resuspended in a buffer appropriate for sequential experiments. To isolate highly purified EVs, density gradient UC is performed. A density gradient is generated using gradient materials, such as sucrose and iodixanol [42,47]. EVs travel to the isopycnic point during centrifugation, and an aliquot collected from the fraction of interest is subjected to additional UC to collect EVs as pellets [42]. While there are several reports claiming that EVs isolated using differential UC contain substantial contaminants, density gradient UC has been demonstrated to isolate EVs of exceptional purity [46,48,49]. Size-based approaches have been reported to co-isolate similar-sized non-vesicular contaminants [48]. Precipitation kits produce a high yield of EVs with low purity [48]. Recent research has described immunoaffinity-based capture and microfluidics-based isolation techniques that permit the isolation of EVs with high purity and yield [42,48]. As there is no consensus on the most effective method for isolation, some publications suggest combining the two approaches [50,51].

Since EVs vary in size and composition, it is necessary to characterize isolated vesicles prior to analysis. Western blots of marker proteins, electron microscopy, and the use of nanoparticle tracking systems are three of the most common approaches to the characterization of EVs. Tetraspanins (such as CD9, CD63, and CD81), MVB-related proteins (such as Alix and TSG101), and heat shock proteins (Hsp70 and Hsp80) are enriched in exosomes, whereas integrins, selectins, and CD40 are enriched in microvesicles [30,31,32,38]. To validate the existence of EVs of interest, Western blotting of marker proteins is carried out. As protein markers specific to RCC-derived EVs, Himbert et al. identified CD147, CD70, and carbonic anhydrase IX(CA IX) [52]. Vesicle size and shape are routinely observed using transmission electron microscopy (TEM) analysis. Additionally, immunoelectron microscopic analysis using a marker protein antibody allows the location of a specific protein in an EV to be visualized [53]. The nanoparticle tracking system provides information on the size distribution and concentration of vesicles. The International Society of Extracellular Vesicles (ISEV) recommends that isolated vesicles be characterized using two or more methods [41].

### 2.3. EV Cargos

EVs modify the phenotypes of recipient cells via the actions of EV cargos such as proteins; RNAs, including mRNA, microRNA (miRNA), and long non-coding RNA (lncRNA); DNA; and lipids [54,55,56,57,58]. Due to the amount of miRNA and protein in EV samples, they have been the subject of intensive research. miRNA is the most abundant RNA species, accounting for 42% of serum exosomal RNA, as demonstrated by sequencing [59]. EV-associated miRNA has been intensively studied and demonstrated to have diverse functions in pathological processes, including in cancer progression [60,61]. Recent studies have shown that miRNAs related to cancer progression are selectively packed into EVs [62]. In addition to miRNA, proteomics studies revealed that various types of proteins are incorporated into EVs isolated from cancer cells [3]. In addition to proteins involved in EV biogenesis and release, such as tetraspanins (CD9, CD63, and CD81) and MVB-related proteins (ALIX and TSG101), proteins associated with cancer progression, including vascular endothelial growth factor (VEGF) and tumor necrosis factor-a (TNF-a), have also been identified by proteomics analyses [63,64].

## 3. Roles of EVs in RCC Progression

In the process of cancer progression, cancer cells acquire essential biological properties, including angiogenesis, host immune system modulation, and metastasis. Additionally, the development of drug resistance is a serious challenge in cancer treatment. Recent research has revealed that cancer-derived EVs are partially responsible for these biological processes. Here, we review EV-mediated angiogenesis, immune system modulation, metastasis, and drug resistance in RCC (Figure 1).

### 3.1. Angiogenesis in RCC and EVs

ccRCC is the most common type of RCC and typically presents with hypervascular tumors. ccRCC is genetically characterized by the inactivation of the von Hippel–Lindau (VHL) gene. The VHL gene is located on chromosome 3p25 and regulates hypoxia response pathways [65]. The VHL protein (pVHL) is a component of the E3 ubiquitin ligase complex that targets hypoxia-inducible factor 1a and 2a (HIF-1a and HIF-2a) for polyubiquitylation and subsequent proteasomal degradation. HIF-1a and HIF-2a are transcription factors that bind to the hypoxia response element (HRE), activating a myriad of genes involved in hypoxia adaptation [66]. In most cases of ccRCC, pVHL is inactivated, resulting in the accumulation of HIF-1a and HIF-2a and continual activation of HIF target genes, despite normal oxygen levels [67,68]. One of the most common targets of HIF1 is vascular endothelial growth factor (VEGF), which induces angiogenesis. Therefore, tyrosine kinase inhibitors (TKIs), which inhibit the VEGF/VEGF receptor signal pathway, are frequently used for anti-angiogenic therapy in clinical practice. However, the response rate ranges from 10% to 31%, indicating that a substantial proportion of ccRCC cases show primary resistance to TKIs [69,70,71,72,73,74]. In addition to VHL loss of function, miRNAs reportedly contribute to VHL inactivation. EVs secreted from RCC contain miR-92a, which targets VHL mRNA. Valera et al. found that miR-92a expression levels are inversely correlated with VHL mRNA levels in ccRCC tissue and concluded that miR-92a regulates the expression level of VHL through mRNA silencing [75,76].

Besides the classical VHL/HIF/VEGF pathway, recent studies have revealed that ccRCC secretes EVs that promote angiogenesis. Horie et al. demonstrated that exosomal carbonic anhydrase 9 (CA Ⅸ) released from RCC cell lines promotes angiogenesis in vitro and that the levels of CA Ⅸ in exosomes are elevated by hypoxic treatment. CA Ⅸ expression is regulated by HIF1 in response to hypoxia [77]. Grange et al. reported that RCC cells expressing CD105, a popular stemness marker, release microvesicles that induce angiogenesis through the function of miRNA cargo, including miR-92a [76]. The miRNA cargo in RCC-derived EVs induces angiogenesis by silencing mRNAs other than of the VHL gene. Hou et al. reported that exosomes secreted from ccRCC cells promote angiogenesis through miR-27a, which targets secreted frizzled-related protein 1 (SFRP1) mRNA [78], and increased expression of SFRP1 induces angiogenesis [79,80]. In addition to these studies, we found that EVs secreted from bone metastatic RCC facilitate angiogenesis and endothelial gap formation in bone marrow in a time-dependent manner and that the angiogenesis observed in our study was partially mediated by aminopeptidase N (APN) located in the EV plasma membrane [81].

RCC-derived EVs partially mediate angiogenesis, which is one of the most distinct characteristics of RCC. Given that inhibiting angiogenesis using TKIs is a mainstay of RCC treatment, EVs involved in angiogenesis could be a promising treatment target.

### 3.2. The Role of EVs in Modulation of the Host Immune System

In the 1950s, the concept of cancer immunosurveillance was proposed by M.F. Burnet. He suggested that the human immune system recognizes cancer cells as non-self due to cancer antigens displayed on the surfaces of antigen-presenting cells (APCs), contributing to the suppression of tumor development [82,83]. Classically, APCs include dendritic cells, macrophages, Langerhans cells, and B cells. In 1996, Raposo et al. first described B cells releasing EVs containing MHC class Ⅱ [84]. Using immunoelectron microscopy, they observed that intraluminal vesicles containing MHC class Ⅱ were released upon fusion with the plasma membrane. Moreover, they found that EVs released from human and murine B lymphocytes induced an antigen-specific MHC class Ⅱ restricted T cell response [84]. More recently, Schreiber RD et al. developed the concept of cancer immunoediting, which refers to the process in which immunity promotes cancer progression as well as eradicates cancer cells [85]. Cancer immunoediting consists of three processes: elimination, equilibrium, and escape. In the elimination phase, immunity suppresses nascent tumor growth. In the equilibrium phase, tumor cells start to evade antitumor immunity due to the lower immunogenicity induced by accumulating mutations. However, in this state, tumor outgrowth does not occur because the balance between tumor growth and tumor elimination is maintained by immune cells [86]. In the escape phase, tumor cells can evade immune surveillance. In this phase, intercellular communications between tumor and immune cells that are mediated by EVs play critical roles [87]. There are some studies describing RCC-derived EVs contributing to immune escape. Grange et al. stated that EVs secreted from RCC cells, especially CD105^+^ RCC cells, inhibit DC maturation and the T cell immune response through the function of HLA g [88]. Moreover, they demonstrated that HLA g blockade leads to the restoration of DC differentiation. Macrophage polarization is one of the critical changes in the tumor microenvironment that favors cancer progression [89]. RCC cells release EVs, promoting a shift in the macrophage subpopulation from M1 macrophages, which suppress tumor proliferation, to M2 macrophages, which suppress the antitumor immune response [90]. It is widely known that tumor cells express PD-L1, which binds to PD-1 on T cell surfaces, inducing immune suppression and leading to cancer progression [91]. Intriguingly, Chen et al. first discovered in 2018 that EVs secreted from malignant melanoma carry PD-L1 on their membrane surfaces [92]. Recent studies have demonstrated that various cancer types secrete EVs with PD-L1 on their surfaces [93]. To the best of our knowledge, there are no studies demonstrating the function of PD-L1 in RCC-derived EVs. However, Qin et al. found that miR-224-5p in EVs contributes to the stability of PD-L1 in RCC cells, suggesting that RCC-derived EVs can induce immune evasion [94]. Xu et al. showed that RCC-derived EVs induce immunosuppression through the inhibition of T cell proliferation. They found that Fas ligands on the surfaces of RCC-derived EVs are responsible for activating the apoptotic pathway in T cells [95]. Diao et al. described the potential effect of RCC-derived EVs on myeloid-derived suppressor cells (MDSCs), which have immune-suppressive effects on adaptive immune responses in cancer. They demonstrated that heat shock protein 70 (Hsp70) packed in EVs secreted from RenCa cells, a murine renal cancer cell line, mediate immune suppression by triggering Stat3 phosphorylation in MDSC [96]. Together, RCC cells secrete EVs that cause immune suppression, suggesting that EVs are possibly involved in the process underlying the acquisition of resistance to immunotherapy. It has been reported that EVs secreted from melanoma cells expressing PD-L1 (PD-L1^+^ EVs) induce immunosuppression by targeting PD1^+^CD8^+^ T cells. PD1^+^CD8^+^ T cells secrete IFN-g, which induces upregulation of PD-L1 levels on EVs [97]. This result suggests that PD-L1^+^ EVs could impact the efficacy of ICI. As far as we know, there are no published studies describing the impact of PD-L1^+^ EVs secreted from RCC on the efficacy of ICI treatments. However, considering that immunotherapy has been a common treatment option for metastatic RCC, further research on cancer immunoediting is exceptionally significant in RCC. Thus, elucidation of the roles played by RCC-derived EVs in this process will provide new insights into the development of novel therapeutic strategies.

### 3.3. Roles of EVs in RCC Metastasis

Given the poor clinical outcomes of mRCC patients, deciphering the mechanism underlying metastasis formation is of great significance [98].

It is well known that the formation of a premetastatic niche, a microenvironment favoring circulating tumor cell attachment, colonization, and growth, is a crucial step in metastasis formation [99]. In this process, EVs secreted from metastatic cancer cells play essential roles in communicating with stromal cells at the metastatic site [100]. Mesenchymal stem cells (MSCs) are also a key component of the metastatic niche [101]. Lindoso et al. found that cancer stem cells (CSC) of RCC release EVs, inducing phenotypic changes in MSCs and resulting in tumor progression [102].

The process of tumor cell dissemination from the primary site to distant organs involves five steps: local invasion, intravasation, surviving in circulation, extravasation, and colonization at the metastatic site [103]. In the initial step of metastasis, tumor cells detach from the primary site and invade surrounding tissue. This process involves the epithelial–mesenchymal transition (EMT) [104]. Jin et al. found that RCC-derived EVs promote cancer cell migration, invasion, and lung metastasis by shuttling metastasis-associated lung adenocarcinoma transcript 1 (MALAT1), a lncRNA that is well known to facilitate cancer metastasis [105]. Wang et al. demonstrated that CSCs of ccRCC release EVs that induce EMT via the function of miR-19b-3p, with the potential to facilitate lung metastasis [106]. Intravasation refers to the step during which tumor cells enter the circulation by crossing the endothelium. Extravasation is another key process in cancer metastasis, where circulating tumor cells break through the barriers of endothelial cells to migrate into stroma at metastatic sites [103]. The enhanced vascular permeability caused by a damaged vascular endothelium is involved in these processes [107]. Jingushi et al. performed proteomic analysis of EVs released from RCC tissue, identifying azurocidin 1 (AZU1) as a functional protein that is comparatively enriched in EVs from RCC tissue than from neighboring normal tissue. In their study, RCC-derived EVs induced an enhancement in vascular permeability through the function of AZU1 [108]. Xuan et al. found decreased levels of miR-549a in the EVs secreted from TKI-resistant RCC cells. miR-549a in EVs regulates HIF-1a expression in vascular endothelial cells, contributing to metastasis by promoting angiogenesis and enhancing vascular permeability [109].

In summary, recent research has revealed that RCC-derived EVs are involved in every step of cancer metastasis.

### 3.4. Roles of EVs in RCC Drug Resistance

In the last two decades, newly developed treatment options for RCC, including targeted therapy and immunotherapy, have contributed to improved clinical outcomes [23]. Therefore, it is critically important to clarify the molecular basis for resistance to these drugs in RCC to contribute to the development of novel biomarkers of responses and the identification of therapeutic targets.

Some researchers have reported differences in EV cargo between resistant and sensitive cells, suggesting that RCC-derived EVs could play roles in drug resistance [110]. Xuan et al. demonstrated that there is decreased miR-549a cargo of the EVs secreted from TKI-resistant RCC cells compared with those from TKI-naïve RCC cells [109]. Moreover, EVs secreted from resistant RCC cells carry biomolecules contributing to cell survival and proliferation during anticancer drug treatments. Zhang et al. indicated that EVs isolated from the culture supernatant of drug-resistant RCC cells contain abundant STAT3 compared with EVs secreted from sensitive cells. They concluded that resistant RCC cells generate EVs that promote cell survival and proliferation in the presence of anticancer drugs by activating the mTOR–ERK–STAT–NF-kb signaling pathway [90]. Qu et al. demonstrated that EVs secreted from sunitinib-resistant RCC cells disseminated sunitinib resistance to sensitive cells through the function of lncRNA, which activates AXL and c-MET signals via competitive binding to miR-34/miR-449 [110]. According to a recent study by Pan et al., RCC-derived EVs contribute to sunitinib resistance by transferring insulin growth factor-like family member 2 antisense 1 (IGFL2-AS1), a novel lncRNA inducing enhanced autophagy [111]. Together, EVs transfer drug resistance from resistant cells to sensitive cells by carrying biomolecules that allow sensitive cells to survive and proliferate during antitumor treatments. This phenomenon is known as horizontal transfer. While the mechanisms of sunitinib resistance induced by RCC-derived EVs have been extensively investigated, the relationship between RCC-derived EVs and resistance to other types of TKI remains unknown. Ishibashi et al. demonstrated in vitro that RCC cells exposed to sunitinib, sorafenib, or pazopanib increased their secretion of interleukin 6, which is thought to play a key role in the development of TKI resistance [112]. Their findings imply that these drugs may develop TKI cross-resistance. Therefore, future research should concentrate on EV-related mechanisms of TKI cross-resistance.

Compared to the classical immunotherapy era, the current standard care for advanced or metastatic RCC, such as TKI, ICI, and their combination, has shown better clinical outcomes, with a response rate of 42–71% [113]. However, a substantial proportion of patients show primary or acquired resistance to these treatments. Therefore, deciphering the mechanisms underlying drug resistance to current immunotherapy is paramount for improving clinical outcomes.

### 3.5. Potential of EV-Targeting Treatment

EVs are potential cancer therapeutic targets due to their involvement in multiple pathways that lead to cancer progression. Several substances that prevent EV formation and release have been discovered in recent studies [114]. Calpeptin, manumycin A, and Y27632 are compounds that reportedly inhibit EV formation. Calpeptin targets calpains, which are calcium-dependent cysteine proteases that are primarily responsible for microvesicle formation. Accordingly, calpeptin reduces the volume of microvesicles that cells secrete [114,115]. Manumycin A inhibits small GTPases from the Ras superfamily that are involved in exosome production and release. Ras regulates multiple cell functions, such as cell differentiation, cell proliferation, adhesion, migration, cytoskeletal integrity, apoptosis, and exosome release. Manumycin A, in combination with GW4869, which is known as an nSMase inhibitor, resulted in a further reduction in exosome release [114,116]. Y27632 decreases microvesicle production and release by inhibiting a Rho-associated protein kinase (ROCK) that mediates signals acting on the cytoskeleton. These agents are proven to reduce EV secretion in prostate cancer, breast cancer, and ovarian cancer cells [114]. As far as we know, no published studies are reporting on the inhibition of EV secretion from RCC cells by these agents. In addition, some compounds suppress EV release by affecting lipid metabolism [114]. Since these agents affect physiological cellular functions, further investigations to determine potential adverse effects are essential in relation to clinical application.

## 4. The Potential of EVs in Clinical Application

### 4.1. The Potential of EVs as Novel Biomarkers for RCC

Currently, cases of RCC are mostly asymptomatic at the initial diagnosis and are identified using computed tomography (CT) or ultrasonography performed for other purposes [117]. However, around 30% of RCC cases present with metastatic disease at the initial diagnosis, and 40% of patients die from RCC progression [118]. This could be partially due to the lack of effective biomarkers. Therefore, novel biomarkers that are useful for early detection or predicting the prognosis of RCC in clinical practice are urgently desired.

Since the characteristics of cancer cells are reflected in EV components, EVs have great potential as a source of real-time information about cancer cells [119]. Several kinds of biomolecules packed in EVs are shown to be promising biomarker candidates for the prediction of prognoses and responses to treatment [120]. In particular, miRNA, a kind of non-coding RNA that is 19–25 nucleotides long and regulates gene expression through the degradation of target mRNA, has been widely investigated as EV cargo. As a biomarker, the presence of miRNA in EVs is crucial, since Evs are thought to protect miRNA, which is otherwise normally degraded in circulation [17]. In the field of RCC biomarker research, Evs isolated from patient plasma, serum, and urine have been thoroughly investigated [43] (Figure 2).

Fujii et al. found that higher levels of miR-224 in EVs isolated from the serum of RCC patients were associated with lower progression-free survival, cancer-specific survival, and overall survival. In an in vitro analysis, the miR-224 of EVs secreted from metastatic RCC cells exhibited increased viability and invasive ability [121]. However, elevated EV levels of miR-224 are not specific to RCC. miR-224 is demonstrated to function as an oncogenic miRNA in several types of cancer, including hepatocellular carcinoma, non-small-cell lung cancer, and colorectal cancer [122,123]. Du et al. found that the miR-let-7i-5p, miR-26a-1-3p, and miR-615-3p levels in plasma EVs are associated with the overall survival (OS) of mRCC patients. In particular, EV-miR-let-7i-5p is strongly predictive of OS in combination with the Memorial Sloan Kettering Cancer Center (MSKCC) score, which is a commonly used risk stratification model for metastatic RCC [124]. Zhang et al. stated that the miR-1233 and miR-210 levels in serum EVs were significantly higher in RCC patients than in healthy controls. Interestingly, the EV-miR-1233 and EV-miR-210 levels were reduced after nephrectomy [125]. Whether considered individually or in combination, miRNAs in EVs have considerable potential as biomarkers for RCC diagnosis, progression, or prognosis prediction.

Urinary EVs are another promising biomarker candidate that could contribute to the early detection and prediction of the prognosis of RCC. Butz et al. reported that RCC patients could be distinguished from healthy volunteers based on urinary EV-miR-126-3p combined with EV-miR-449a or EV-miR-34b-5p [126]. Urine collected from patients with RCC undergoing nephrectomy showed higher levels of EV-miR-210-3p than healthy controls. In addition, the urinary EV-miR-210-3p levels were reduced after surgical resection, suggesting that urinary EV-miR-210-3p could be a useful biomarker for post-nephrectomy follow-up [127]. Other RNA types, such as lncRNA and mRNA, have also been reported as potential biomarkers [128].

Recent advances in proteomics have allowed researchers to explore promising protein biomarkers in EV components. Jingushi et al. identified 3871 proteins in EVs isolated from the culture supernatant of RCC tissue and the surrounding non-tumor tissue. Of these, azurocidin (AZU1) was shown to be enriched in EVs secreted from tumor tissue. Additionally, they found that the EV-AZU1 content was significantly higher in patients with RCC than in healthy controls. Their findings showed the possibility of EV-AZU1 as a novel biomarker for RCC [108]. Zhao et al. found that polymerase I and transcript release factor (PTRF)/Cavin1 in urinary EVs could be a promising biomarkers for ccRCC diagnosis. According to their data, the PTRF level was significantly higher in urinary EVs collected from RCC patients than those from healthy volunteers and was also reduced after surgery [129]. Horie et al. found that γ-glutamyltransferase levels in serum EVs are elevated in patients with advanced RCC pathology [130]. We discovered that APN levels in extracellular vesicles (EVs) secreted from tissue samples are higher in RCC patients with bone metastases than in patients without metastasis [81]. Although no specific EV-based biomarker has been implicated in RCC clinical practice to date, a rapidly expanding number of biomolecules have been identified in RCC-derived EVs, and miRNAs, lncRNA, and proteins in EVs are promising candidate biomarkers for RCC (Table 1).

### 4.2. EVs as Novel Drug Carriers

Despite rapid advances in cancer treatment, the clinical outcome is still unsatisfactory. To achieve the maximum antitumor effect with minimum adverse effects, the development of novel carriers of therapeutic agents is highly desired. As one of the most promising novel drug carrier candidates, there are high expectations of EVs due to their excellent biological properties, including minimal immune reaction and efficient uptake by recipient cells. Recent advances in nanotechnology have allowed researchers to load EVs with antitumor drugs and biomolecules. For instance, EVs loaded with docetaxel (DTX) showed promising outcomes. Wang et al. demonstrated that EVs loaded with DTX showed significant therapeutic effects both in vitro and in vivo. In that study, DTX was packed into EVs collected from a non-small-cell lung cancer cell line via electroporation [131]. Cenik et al. observed that EVs loaded with DTX induce mitochondrial apoptosis in HeLa cells [132]. In addition to DTX, paclitaxel, and doxorubicin also exhibit therapeutic effects as EV cargo [133]. Small RNAs, such as small interfering RNA (siRNA) and microRNA (miRNA), are also commonly packed into EVs for use as antitumor agents. Wang et al. demonstrated that EVs loaded with let-7 miRNA showed antitumor effects in both in vitro and in vivo settings [134]. In the field of RCC treatment, TNF-related apoptosis-inducing ligand (TRAIL) loaded into EVs secreted from mesenchymal stromal cells was demonstrated to induce apoptosis in the renal cancer cell lines RCC10 and HA7-RCC [135]. To enhance therapeutic efficacy, determining the mechanism by which EVs are specifically targeted to cancer cells after systemic injection is necessary. Wang et al. combined EVs with AS1411, a ligand of nucleolin, which is overexpressed on the plasma membranes of breast cancer cells. AS1411-EVs loaded with let-7 miRNA were more specifically targeted to tumors in a CDX mouse model than EVs containing let-7 alone, i.e., without AS1411. This result suggests that ligands of cancer-specific biomolecules located on the plasma membrane have great potential to contribute to the specific distribution of EVs to cancer cells [134]. As far as we know, few publications demonstrate the efficacy of EVs as a drug delivery system in RCC treatment. However, G250, which is expressed in 85% of RCC cells but is rarely expressed in neighboring normal tissue, could be a promising target for RCC-specific drug delivery [136].

### 4.3. EVs as Biological Response Modifiers

EVs are attracting researchers’ attention not only as a drug delivery system but also as a biological response modifier (BRM). In the field of cancer treatment, the term ‘BRM’ refers to agents that boost the immune response against tumor cells. Interferon α and interleukin-2 were previously used as BRM in treating RCC, with unsatisfactory clinical outcomes [137]. Since EVs derived from immune cells can stimulate the host immune response, they are considered promising BRM. EVs released from dendritic cells which are one of the most powerful antigen-presenting cells stimulate CD4^+^ and CD8^+^ T cell immunity through the function of MHC class I, MHC class II, CD86, and Hsp 70–90 chaperons [138]. To effectively eliminate cancer cells, cytotoxic T lymphocyte (CTL) activation is required. Wu et al. found that fully activated CTL secretes EVs that stimulate bystander CTL in the presence of IL-12 [139]. These findings suggest that EVs secreted from immune cells may act as BRM. Since RCC shows favorable responses to current immunotherapies, such as immune checkpoint inhibitors, rather than chemotherapy, intensive efforts have been made to explore novel BRMs that enhance the therapeutic power of cancer immunotherapy for RCC. Zhang et al. discovered that EVs secreted from IL-12-anchored RCC cells promote proliferation and IFN g release from T cells, leading to significant in vitro cytotoxic effects [140]. Xu et al. demonstrated that EVs secreted from RenCa cells, a murine kidney cancer cell line, enhance the CD8^+^ T cell-mediated antitumor response via the Fas ligand/Fas signaling pathway [95]. These findings indicate that EVs are a prospective BRM for immunotherapy of RCC.

### 4.4. Limitations for the Clinical Application of EVs as a Therapeutic Modality

The main challenge for the clinical application of EV-based cancer treatment lies in the efficacy of EV production and cargo loading. To overcome this challenge, it is necessary to develop novel methods for efficiently producing EVs and loading them with therapeutic agents. In the majority of studies exploring EV-based cancer treatment, the source of EVs is cancer cells or immune cells (e.g., dendritic cells, macrophages, and T cells) [138]. The main advantage of using cancer-derived EVs for EV-based drug delivery is that their distribution is specific to the parent cells. This biological property allows for efficiently delivering antitumor agents to cancer cells. However, many studies have indicated the involvement of cancer-derived EVs in various processes of cancer progression, such as drug resistance and metastatic niche formation, and immune suppression, and there is a risk of promoting cancer progression when using cancer-derived EVs [141].

Immune cells are frequently used as a source of EVs in EV-based immunotherapy. In particular, EVs secreted from dendritic cells, which play a critical role in antigen presentation and T cell priming, are frequently used as BRMs in EV-based cancer immunotherapy. Dendritic-cell-derived EVs preserve the powerful antigen presentation capacity of their parent cells, stimulating antitumor responses. Stem cells, especially mesenchymal stem cells, are also recognized as an ideal source of EVs for clinical application due to their high proliferative capacity and low immunogenicity. The intrinsic cancer tropism of EVs derived from MSCs reportedly supports the specific distribution of EV load to cancer cells [142]. However, numerous studies indicate that MSCs and EVs derived from MSCs either promote or inhibit cancer progression [142]. Therefore, further study is necessary for the clinical application of EVs derived from MSCs. In addition to EVs secreted from these cells, EVs included in bovine milk are attracting researchers’ attention as a promising source of EVs due to their cost, safety, and mass producibility. Bovine-milk-derived EVs have shown excellent biological stability in low-pH conditions in the stomach, allowing oral intake [143,144]. Plant-derived EVs are also investigated as promising drug carriers for cancer treatment. EVs derived from edible plants, such as citrus and lemon, are extracted. The advantages of using EVs derived from plants are their mass production and biological stability, allowing for oral administration [145]. In the clinical application of EV-based cancer treatment, developing an effective technique for cargo loading is a crucial step. Methods for cargo loading include pre-loading, which describes the treatment of donor cells, and post-loading, which describes the treatment of EVs. In pre-loading, genetic manipulation and co-incubation are frequently used. The overexpression of target genes in donor cells permits an increase in the number of target molecules loaded onto EVs. Another method is incubating therapeutic cargo with donor cells. During co-incubation, donor cells are incubated with cargo until they are absorbed by donor cells. EVs loaded with cargo, which are secreted into the supernatant, are then isolated. Low loading efficiency and an extended procedure are two disadvantages of pre-loading. The loading efficacy in co-incubation and genetic manipulation is typically low and dependent on the therapeutic agent and donor cell characteristics [143]. Electroporation, sonication, transfection, and co-incubation are well-known post-loading techniques. During electroporation, electrical pulses create temporal pores through which small RNAs, DNAs, drugs, and chemicals are incorporated into EVs. There has been a recent increase in the efficacy of electroporation for EV cargo loading as a result of the development of an optimized protocol; however, it may cause RNA aggregation when loading RNAs [143]. Sonication is another common active cargo loading technique. It demonstrated a high loading efficacy. The disadvantages of sonication include the destruction of the EV membrane and its cargo. Lipofection is utilized to efficiently insert nucleic acids into EVs. UC isolates transfection complexes along with EVs, which can influence EV function and uptake. During co-incubation, therapeutic agents passively diffuse into EVs through interaction with the lipid bilayer membrane. Co-incubation is commonly employed because of its simple process. However, the loading efficacy is insufficient, and further investigation is required to implement improvements.

## 5. Conclusions

Recent research has shown that EVs are involved in various types of cancer progression. Current evidence indicates that RCC-derived EVs contribute to RCC progression through host immune modulation, drug resistance, and metastasis. EVs are therefore considered a prospective biomarker and therapeutic target for RCC. However, several limitations must be resolved in the clinical application of EV-based diagnosis and therapeutics.

## Figures and Tables

**Figure 1 nanomaterials-13-01611-f001:**
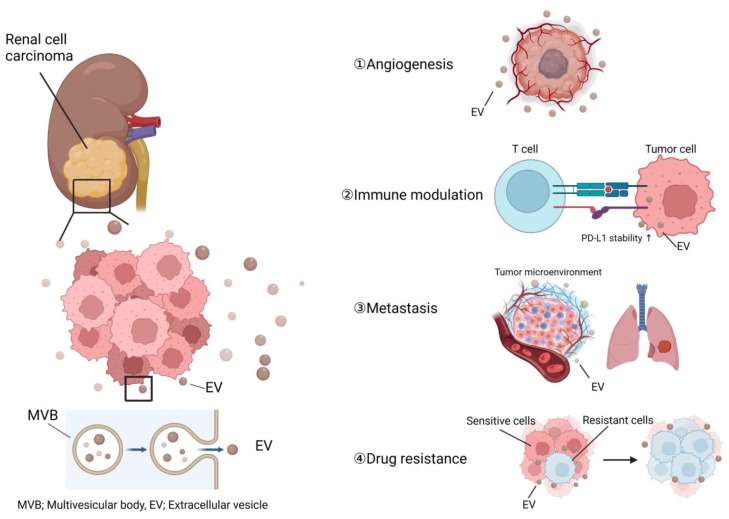
The major roles of EVs in RCC progression. EVs secreted from RCC cells contribute to cancer progression through angiogenesis, host immune modulation, metastasis, and drug resistance. MVB, multivesicular body.

**Figure 2 nanomaterials-13-01611-f002:**
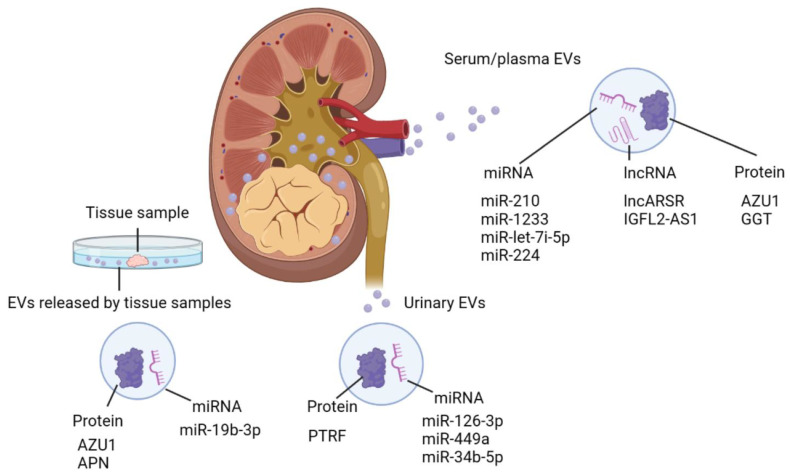
Biomolecules identified in Evs isolated from RCC clinical samples. Evs isolated from clinical samples of RCC carry various biomolecules, including miRNA, lncRNA and protein. Many of these contribute to RCC progression. ARSR, lncRNA Activated in RCC with Sunitinib Resistance; APN, aminopeptidase N; Azu1, azurocidin; PTRF, polymerase I, and transcript release factor; GGT, γ-glutamyltransferase; IGFL2-AS1, insulin growth factor-like family member 2 antisense 1.

**Table 1 nanomaterials-13-01611-t001:** Biomolecules identified in EVs isolated from RCC clinical samples and cell lines. EVs isolated from clinical samples and cell lines of RCC carry various biomolecules, including miRNA, lncRNA, and protein. Some of them have been analyzed for their function. ARSR, lncRNA Activated in RCC with Sunitinib Resistance; APN, aminopeptidase N; Azu1, azurocidin; PTRF, polymerase I and transcript release factor; GGT, γ-glutamyltransferase; IGFL2-AS1, insulin growth factor-like family member 2 antisense 1.EMT, epithelial-mesenchymal transition.

EV Source	Cargo Type	Cargo Specific	Function/Application	Reference
Serum/plasma	miRNA	miR-210	Biomarker	[125]
		miR-1233	Biomarker	[125]
		miR-224-5p	Stability of PD-L1 in RCC	[94]
		miR-let-7i-5p	Biomarker	[124]
	lncRNA	lncARSR	Sunitinib resistance	[110]
		IGFL2-AS1	Autophagy, Sunitinib resistance	[111]
	protein	AZU1	Elevation in vascular permeability, Biomarker	[108]
		GGT	Microvascular invasion	[130]
Urine	miRNA	miR-126-3p	Biomarker	[126]
		miR-449a	Biomarker	[126]
		miR-34b-5p	Biomarker	[126]
	protein	PTRF	Biomarker	[129]
Tissue	protein	AZU1	Elevation in vascular permeability	[108]
		APN	Angiogenesis	[81]
	miRNA	miR-19b-3p	EMT	[106]
Cell line	miRNA	miR-27a	Angiogenesis	[78]
		miR-92a	Angiogenesis	[76]
		miR-19b-3p	EMT	[106]
	lncRNA	MALAT1	Proliferation	[105]
			Lung metastasis formation	
		lncARSR	Macrophage polarization	[90]
	Protein	CAⅨ	Angiogenesis	[77]
		HLA-G	Inhibition of DC differentiation and T cell immune response	[88]
		APN	Angiogenesis	[81]

## Data Availability

Not applicable.

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
