# Peer review of "The Roles of Extracellular Vesicles in the Progression of Renal Cell Carcinoma and Their Potential for Future Clinical Application"

_nanomaterials, 2023, doi:10.3390/nano13101611_

Round 1
Reviewer 1 Report
The manuscript gathers the newest data on extracellular vesicles role in cancer, with focus on renal carcinoma, in order to establish novel therapeutically approaches.
The authors present an essential introduction on renal cell carcinoma, a straightforward characterization of different types of extracellular vesicles, and then, step by step, the authors display the multiple implications of EVs in RCC tumor progression, antitumor immune response and how EV influences the therapy efficacy.
The manuscript revealed how EVs could became prognostic factors or targets, but also they disclose the limitations of EVs clinical use.
I recommend accepting the paper, after minor revisions, as follows:
Abstract: the EV abbreviation has to be explained in the abstract. Please revise: "especially as biomarkers and a therapeutic modality"
As well, I suggest explaining all abbreviations in the figure caption (Figures 1 and 2).
Introduction, page 1: "after treatment with curative intent [21,22]" – a brief description of the standard treatments in RCC is needed in this paragraph
Page 2, chapter 2.1 Which EV type is characteristic to RCC? The described features (CD9, CD40, CD63, CD 81, Alix) are present in RCC-derived EV as well?
Page 2, chapter 2.2. For the readers who are not familiar with the experiments described here, it has to be stated what kind of samples are subjected to centrifugation (tissue, biological fluids, cell culture supernatant, or other) and if they are specific challenges in EV isolation from samples from RCC patients- between references 42-51 it is only one study on RCC.
Page 6, chapter 3.4 – some data are needed regarding the drugs which trigger the MDR mechanisms in RCC, others that sunitinib. The last paragraph of this chapter: “EVs are possibly involved in the process underlying acquisition (.......) fits well to 3.2.
Page 9, chapter 4.4. please specify how the cited studies recommend to use cancer-derived EVs. The immune cells, sources of EV, were from cancer patients of from healthy donors?
Author Response
Abstract: the EV abbreviation has to be explained in the abstract. Please revise: "especially as biomarkers and a therapeutic modality"
As well, I suggest explaining all abbreviations in the figure caption (Figures 1 and 2).
We really thank for the reviewer’s kind suggestion. We have added EV abbreviation in the abstract and deleted the phrase ‘especially as biomarkers and a therapeutic modality’. In addition, all the abbreviations are included in Figure 1 and 2.
Introduction, page 1: "after treatment with curative intent [21,22]" – a brief description of the standard treatments in RCC is needed in this paragraph
Thank you so much for your kind comment. We have added the sentence ‘Localized kidney cancer is typically treated with surgery, whereas metastatic kidney cancer is typically treated with systemic therapy [23].’ in this paragraph.
Page 2, chapter 2.1 Which EV type is characteristic to RCC? The described features (CD9, CD40, CD63, CD 81, Alix) are present in RCC-derived EV as well?
Exosomes are the most extensively researched EV type across all cancer types, including RCC. Tetraspanins (CD9, CD63 and CD81) and MVB-related proteins (TSG101 and Alix) are enriched in majority of EVs including RCC-derived EVs. In many studies of RCC-derived EVs, these proteins are used to confirm the isolation of qualified EVs.
Page 2, chapter 2.2. For the readers who are not familiar with the experiments described here, it has to be stated what kind of samples are subjected to centrifugation (tissue, biological fluids, cell culture supernatant, or other) and if they are specific challenges in EV isolation from samples from RCC patients- between references 42-51 it is only one study on RCC.
We sincerely thank to the reviewer’s thoughtful comment. Differential UC is employed to isolate EVs from various samples including tissues, biological fluids and supernatants. To clarify this point, we have added the sentence’ Of these, differential centrifugation involving UC is the most traditional and widely accepted method and employed in EV isolation from various sample types such as biological fluids and cell culture supernatants.’ in this section. The difficulties in isolating EVs are not peculiar to EVs derived from RCC. It is nearly universal among EV researchers.
Page 6, chapter 3.4 – some data are needed regarding the drugs which trigger the MDR mechanisms in RCC, others that sunitinib. The last paragraph of this chapter: “EVs are possibly involved in the process underlying acquisition (.......) fits well to 3.2.
We really thank to the reviewer’s kind suggestion. To discuss MDR in RCC more, we have added the sentence ’ While the mechanisms of sunitinib resistance induced by RCC-derived EVs have been extensively investigated, the relationship between RCC-derived EVs and resistance to other types of TKI remains unknown. Ishibashi et al. demonstrated in vitro that RCC cells exposed to sunitinib, sorafenib, or pazopanib increased their secretion of interleukin 6, which is thought to play a key role in the development of TKI resistance [112]. Their findings imply that these drugs may develop TKI cross-resistance. Therefore, future research should concentrate on EV-related mechanisms of TKI cross-resistance. ‘ in page page6, the section 3.4.
We have moved the last paragraph to the section 3.2.
Page 9, chapter 4.4. please specify how the cited studies recommend to use cancer-derived EVs. The immune cells, sources of EV, were from cancer patients of from healthy donors?
Cancer cells produce EVs significantly more efficiently than non-cancerous cells, and cancer derived EVs are selectively absorbed by cancer cells, according to cited studies. In studies employing a preclinical model, EVs derived from mouse immune cells are utilized.
Reviewer 2 Report
This manuscript reviewed the mechanism of disease progression caused by RCC-derived EVs. Authors also summarized the potential clinic applications and challenges of using EVs for diagnostics and therapeutics. Overall, the topic is significant, and the manuscript is well organized. However, additional information is required for a more comprehensive overview.
1. Are there any special characterizations of RCC-derived EVs in comparison with other cells-derived EVs?
2. Are the roles of RCC-derived EVs in RCC progression specifically or other tumor cells-derived EVs also have similar functions?
3. An additional table to summarize the role of various components of RCC-derived EVs in tumor progression is suggested.
4. The different strategies to load drugs into EVs is suggested to summarize systemically, as well as their advantages and disadvantages are recommended to discuss.
5. More examples and discussions are suggested to support that the EVs can serve as biological response modifiers.
6. The risks of using tumor cells-derived EVs are suggested to address.
7. For the clinical application of EV-based cancer treatment, EVs derived from stem cells and plants are recommend discussing.
Minor editing of English language required.
Author Response
- Are there any special characterizations of RCC-derived EVs in comparison with other cells-derived EVs?
CD147, CA9 and CD70 has been reported as specific biomarkers of EVs derived from clear cell renal cell carcinoma. We added the sentense’ As protein markers specific to RCC-derived EVs, Himbert et al. identified CD147, CD70 and carbonic anhydrase IX(CA IX) [52].’ in page3, section 2.2.
- Are the roles of RCC-derived EVs in RCC progression specifically or other tumor cells-derived EVs also have similar functions?
The roles of RCC-derived EVs are similar to those of other tumor cells-derived EVs. Among several functions of EVs, more studies are focused on angiogenesis in RCC because clinical RCC is characterized by hypervascularity (Table1).
- An additional table to summarize the role of various components of RCC-derived EVs in tumor progression is suggested.
We sincerely appreciate the reviewer’s thoughtful suggestion. We have created Table1 to summarize various components of RCC-derived EVs.
- The different strategies to load drugs into EVs is suggested to summarize systemically, as well as their advantages and disadvantages are recommended to discuss.
We have revised the section 4.4 (page9-10) to summarize cargo loading strategies.
- More examples and discussions are suggested to support that the EVs can serve as biological response modifiers. 
To provide additional examples of BRM, we have added the paragraph’ In the field of cancer treatment, the term 'BRM' refers to agents that boost the immune response against tumor cells. Interferon a and interleukin-2 were previously used as BRM in the treatment of RCC, with unsatisfactory clinical outcomes [137]. Since EVs derived from immune cells possess the ability to stimulate the host immune response, they are regarded as a promising BRM. EVs released from dendritic cells which are one of the most powerful antigen presenting cells stimulate CD4+ and CD8+ T cell immunity through the function of MHC class I, MHC class II, CD86 and Hsp 70-90 chaperons [138]. To effectively eliminate cancer cells, cytotoxic T lymphocyte (CTL) activation is required. Wu et al. found that fully-activated CTL secretes EVs that stimulate bystander CTL in the presence of IL-12 [139]. These findings suggest that EVs secreted from immune cells may act as BRM.’ in the section 4.3 in page 9.
- The risks of using tumor cells-derived EVs are suggested to address.
To clarify the risk of cancer-derived EVs, we have added the sentence’ However, many studies have indicated the involvement of cancer-derived EVs in various processes of cancer progression, such as drug resistance, metastatic niche formation and immune suppression. Therefore, there is a risk of promoting cancer progression when using cancer-derived EVs [141].’in the section 4.4 in page 10.
- For the clinical application of EV-based cancer treatment, EVs derived from stem cells and plants are recommend discussing.
We discuss the potential of EVs derived from stem cells and plants in the section 4.4 in page 10.